# Analysis of Nutritional Quality Attributes and Their Inter-Relationship in Maize Inbred Lines for Sustainable Livelihood

Sapna Langyan [1,2,*], Zahoor A. Dar [3], D. P. Chaudhary [1], J. C. Shekhar [1], Susila Herlambang [4], Hesham El Enshasy [5,6], R. Z. Sayyed [7,*] and S. Rakshit [1]

[1] ICAR-Indian Institute of Maize Research, PAU Campus, Ludhiana, Punjab 141004, India; chaudharydp@gmail.com (D.P.C.); jcsdmr@yahoo.co.in (J.C.S.); pdmaize@gmail.com (S.R.)
[2] ICAR-National Bureau of Plant Genetic Resources, Pusa Campus, New Delhi 110012, India
[3] Dryland Agriculture Research Station, SKUAST, Kashmir 190001, India; zahoorpbg@gmail.com
[4] Department of Soil Science, Faculty of Agriculture, Pembangunan Nasional Veteran University, Yogyakarta 55293, Indonesia; susilaherlambang@upnyk.ac.id
[5] Institute of Bioproduct Development (IBD), Universiti Teknologi Malaysia (UTM), Skudai, Johor Bahru 81310, Malaysia; henshasy@ibd.utm.my
[6] City of Scientific Research and Technology Applications, New Burg Al Arab, Alexandria 21934, Egypt
[7] Department of Microbiology, PSGVP Mandal's Arts, Science, Commerce College, Shahada 425409, India
* Correspondence: sapna@icar.gov.in (S.L.); sayyedrz@gmail.com (R.Z.S.)

**Abstract:** The present investigation was planned to understand the variability and inter-relationship among various nutritional quality attributes of maize kernels to identify potential donors of the respective traits for future hybridization programs. Sixty-three maize inbred lines were processed for the estimation of protein, starch, fat, sugar, 100-kernel weight, specific gravity, and moisture level of the grain. The results reveal that a wide variability among protein, starch, 100-kernel weight, specific gravity, and fat was seen, with special emphasis on the protein concentration that varied from 8.83 to 15.54%, starch (67.43–75.31%), and 100-kernel weight (9.14–36.11 gm). Factor analysis revealed that the protein concentration, starch, and 100-kernel weight, the three major components, comprise 68.58% of the kernel variability. Protein exhibited a significant negative correlation with starch and 100-kernel weight, indicating that an increase in the protein concentration will down-regulate the starch and 100-kernel weight. The inbred lines are proposed as donors for the development of high cultivars for their respective traits, viz., high protein (DMR WNC NY 403 and DMR WNC NY 404), high starch concentration (DMR WNC NY 2163, DMR WNC NY 2219, DMR WNC NY 2234, DMR WNC NY 2408, DMR WNC NY 2437, and DMR WNC NY 2466), high 100-kernel wt. (DMR WNC NY 2113, DMR WNC NY 2213, DMR WNC NY 2233, DMR WNC NY 2234, DMR WNC NY 2414, DMR WNC NY 2435, DMR WNC NY 2465, and DMR WNC NY 2474), sugar (DMR WNC NY 2417), and specific gravity (DMR WNC NY 2418). Genetic distance analysis revealed that DMR WNC NY 397 and DMR WNC NY 404 are the farthest apart inbred lines, having major differences in their protein, fat, starch, and sugar contents, followed by DMR WNC NY 2436 and DMR WNC NY 2394, DMR WNC NY 2212 and DMR WNC NY 2430, DMR WNC NY 396 and DMR WNC NY 2415, DMR WNC NY 404 and DMR WNC NY 2144, and DMR WNC NY403 and DMR WNC NY 2115. Moreover, this study proposes that these possible combinations of lines (in a breeding program) would result in genetic variability with simultaneous high values for the respective characteristics.

**Keywords:** clustering; correlation; fat; protein; starch; sugar; specific gravity

## 1. Introduction

Food insecurity and hunger affect more than 900 million people worldwide each year. People from poor and underdeveloped and some developing nations are at greater risk. About 5 million hungry people in the world die each year from nutrient deficiency

causes [1,2]. Women and children are more susceptible to nutrient deficiency due to reproduction and growth demands, respectively [3]. Maize (*Zea mays* L.) is one of the world's important cereal grain crops after rice and wheat. The United States, the European Union, China, Brazil, Mexico, and India are the world's leading producers of maize [4]. Its popularity as a crop is largely due to its diverse functionality as a food source for both humans and animals. Maize has been the major source of the world's protein and calories [5,6]; hence, it is the dietary staple food crop for more than 300 million people [7].

India, since independence, has achieved a manifold increase in food grain production owing to the Green Revolution of the late 1960s. Indian agriculture research, until lately, insisted on increasing productivity per se, rather than quality. India's Human Development Index rank of 130 in the year 2020 reflects a major deficiency in the quality of life of people. Various initiatives of the government have largely remained unsuccessful in bringing down the menace of malnutrition. The latest National Family and Health Survey (NHFS2018-19) data released by the government are indicative of the limited progress made in improving the nutritional status and quality of health services for infants and children, adolescent girls, and women [8]. This is because an adequate diet is not affordable and accessible to all; worldwide, 462 million adults are underweight, and around 45% of deaths among children under 5 years of age are linked to undernutrition [9]. India is one of the nations with a large number of tribal people having malnutrition. This warrants an urgent need to produce maize with sufficient nutrient contents, as maize is a staple and principal food crop for the majority of the tribal people [10]. Various strategies to improve nutritional quality include increasing access to various nutrients, food fortification with supplements, bio-fortification, the introduction of pharmaceutical supplements, and dietary diversification [11,12].

Maize (*Zea mays* L.) is the most widely cultivated crop in all conditions, ranging from tropical to temperate, and regions of the world, providing nutrients as well as raw materials for biomolecules such as starch, fat, and protein [13]. In India, maize occupies a prominent position, and each part of the maize plant is utilized in one or another way, with nothing going to waste [14,15]. The utilization pattern of maize comprises 59% as feed, 17% for industrial purposes, 10% as food, around 10% for export, and 4% for other purposes, including seed [16]. Among all cereals, maize has the highest growth rate with maximum productivity, and due to it possessing the highest genetic yield potential, maize is known as the "queen of the cereals" [17]. The nutritional composition of normal maize comprises 8–13% protein, 68–73% starch, 2–5% fat, 2–4% sugar, fibers, minerals, etc. [18,19]. However, these nutritional attributes are inter-related, and an increase in one may adversely affect the other such as high-fat maize likely having a lower quantity of starch [20,21]. Keeping in view the inter-relationship of nutritional quality parameters when using the information in hybrid breeding programs, the present research work was undertaken to study the variability, the correlation, and the inter-relationship between the nutrient components of 63 different maize inbred lines. Additionally, the genetic distance between 63 different varieties of maize inbred lines was studied in order to select those that have the potential to be used as donor parents for their respective traits for the development of maize cultivars with enhanced protein, sugar, fat, and starch, with a possible contribution to decreasing malnutrition.

## 2. Materials and Methods

### 2.1. Plant Materials

The inbred lines were grown in an augmented block design (ABD), with 4 rows per inbred line at 60 cm spacing at a length of 3 m at the Winter Nursery Centre, Hyderabad, India, having a temperature range of 7 to 12 °C, sandy soil, 6.5 pH, and low organic matter, during the rabi season. The plants were selfed, and the selfed seeds were used for the biochemical evaluation. Details of the pedigree of the inbred lines are provided in Table 1.

**Table 1.** Nutritional composition of 63 maize inbred lines.

| PEDIGREE | Protein (%) | Fat (%) | Sugar (%) | Starch (%) | 100-K wt. (g) | Specific Gravity (g/cm³) |
|---|---|---|---|---|---|---|
| DMR WNC NY 396 | 12.39 | 3.08 | 3.58 | 68.46 | 14.3 | 1.19 |
| DMR WNC NY 397 | 11.56 | 3.19 | 3.24 | 70.60 | 17.00 | 1.42 |
| DMR WNC NY 398 | 12.38 | 3.05 | 3.82 | 68.05 | 24.50 | 1.23 |
| DMR WNC NY 399 | 10.31 | 3.23 | 3.71 | 68.44 | 19.90 | 1.67 |
| DMR WNC NY 400 | 9.86 | 4.47 | 3.24 | 73.39 | 27.90 | 1.27 |
| DMR WNC NY 403 | 15.54 | 2.41 | 3.42 | 67.43 | 24.40 | 1.35 |
| DMR WNC NY 404 | 13.52 | 2.48 | 3.55 | 70.22 | 26.70 | 1.34 |
| DMR WNC NY 2430 | 12.20 | 3.56 | 3.08 | 70.29 | 25.00 | 1.39 |
| DMR WNC NY 2392 | 9.83 | 3.15 | 3.44 | 71.33 | 23.54 | 1.47 |
| DMR WNC NY 2393 | 10.45 | 3.28 | 3.14 | 71.93 | 20.70 | 1.15 |
| DMR WNC NY 2394 | 12.07 | 2.57 | 3.56 | 68.80 | 26.30 | 1.10 |
| DMR WNC NY 2395 | 10.61 | 2.21 | 3.47 | 72.90 | 26.80 | 1.41 |
| DMR WNC NY 2396 | 11.24 | 2.26 | 3.38 | 68.33 | 23.50 | 1.10 |
| DMR WNC NY 2397 | 10.72 | 2.36 | 3.90 | 67.93 | 24.65 | 1.17 |
| DMR WNC NY 2398 | 11.00 | 2.92 | 4.27 | 70.55 | 21.37 | 1.19 |
| DMR WNC NY 2399 | 11.50 | 2.87 | 4.04 | 68.22 | 22.60 | 1.26 |
| DMR WNC NY 2431 | 11.19 | 2.64 | 3.23 | 73.07 | 20.66 | 1.15 |
| DMR WNC NY 2430 | 12.51 | 2.89 | 3.08 | 68.85 | 27.90 | 1.27 |
| DMR WNC NY 2400 | 11.86 | 3.36 | 3.65 | 68.91 | 24.00 | 1.26 |
| DMR WNC NY 2401 | 12.61 | 2.50 | 3.51 | 68.42 | 21.76 | 1.21 |
| DMR WNC NY 2402 | 12.06 | 3.22 | 3.06 | 69.04 | 27.10 | 1.13 |
| DMR WNC NY 2403 | 8.83 | 2.43 | 3.74 | 70.79 | 28.30 | 1.18 |
| DMR WNC NY 2404 | 9.79 | 2.32 | 3.86 | 70.47 | 26.70 | 1.21 |
| DMR WNC NY 2405 | 9.75 | 2.58 | 3.28 | 68.96 | 26.30 | 1.20 |
| DMR WNC NY 2432 | 11.83 | 3.30 | 3.33 | 72.64 | 22.30 | 1.17 |
| DMR WNC NY 2433 | 10.42 | 2.29 | 3.13 | 68.22 | 21.20 | 1.12 |
| DMR WNC NY 2406 | 11.91 | 2.56 | 3.01 | 70.42 | 29.30 | 1.13 |
| DMR WNC NY 2407 | 11.79 | 2.28 | 3.64 | 70.63 | 20.05 | 1.22 |
| DMR WNC NY 2408 | 11.12 | 2.69 | 3.28 | 74.92 | 25.80 | 1.17 |
| DMR WNC NY 2434 | 9.55 | 3.05 | 3.65 | 73.35 | 23.70 | 1.25 |
| DMR WNC NY 2409 | 9.76 | 2.28 | 4.34 | 72.66 | 25.8 | 1.17 |
| DMR WNC NY 2410 | 12.03 | 2.23 | 3.71 | 73.18 | 29.90 | 1.15 |
| DMR WNC NY 2435 | 12.50 | 2.73 | 3.68 | 69.78 | 35.60 | 1.19 |
| DMR WNC NY 2436 | 11.31 | 3.23 | 3.75 | 70.73 | 17.40 | 1.16 |
| DMR WNC NY 2412 | 11.65 | 2.61 | 3.10 | 72.51 | 24.40 | 1.22 |
| DMR WNC NY 2414 | 11.21 | 2.04 | 3.01 | 68.92 | 30.90 | 1.14 |
| DMR WNC NY 2415 | 11.24 | 3.22 | 3.45 | 69.25 | 27.30 | 1.14 |
| DMR WNC NY 2416 | 9.64 | 2.72 | 3.56 | 71.35 | 27.00 | 1.17 |
| DMR WNC NY 2417 | 11.35 | 2.84 | 5.37 | 71.91 | 26.40 | 1.20 |
| DMR WNC NY 2418 | 12.67 | 3.50 | 3.68 | 72.75 | 19.05 | 1.90 |
| DMR WNC NY 2419 | 11.56 | 2.35 | 3.17 | 70.30 | 21.79 | 1.21 |
| DMR WNC NY 403 | 15.54 | 2.41 | 3.42 | 67.43 | 9.14 | 0.96 |
| DMR WNC NY 404 | 13.52 | 2.48 | 3.55 | 70.22 | 13.63 | 1.24 |
| DMR WNC NY 2437 | 11.31 | 3.18 | 3.75 | 74.26 | 21.94 | 1.22 |
| DMR WNC NY 2462 | 12.29 | 2.23 | 3.52 | 70.15 | 29.98 | 1.36 |
| DMR WNC NY 2208 | 12.19 | 3.01 | 3.34 | 71.48 | 24.33 | 1.28 |
| DMR WNC NY 2212 | 12.65 | 3.21 | 3.91 | 68.47 | 17.26 | 1.57 |
| DMR WNC NY 2213 | 10.66 | 2.45 | 4.86 | 71.35 | 33.30 | 1.15 |
| DMR WNC NY 2469 | 10.52 | 2.47 | 3.15 | 70.89 | 24.74 | 1.24 |
| DMR WNC NY 2219 | 11.82 | 2.82 | 4.28 | 74.66 | 19.61 | 1.63 |
| DMR WNC NY 2233 | 12.22 | 2.50 | 4.87 | 71.36 | 33.83 | 1.41 |
| DMR WNC NY 2234 | 10.12 | 2.59 | 4.74 | 75.31 | 36.11 | 1.20 |
| DMR WNC NY 2113 | 11.08 | 3.05 | 4.40 | 71.72 | 35.43 | 1.27 |
| DMR WNC NY 2465 | 9.93 | 2.48 | 3.65 | 72.53 | 35.94 | 1.28 |
| DMR WNC NY 2466 | 10.00 | 2.68 | 3.38 | 74.76 | 31.63 | 1.44 |
| DMR WNC NY2138 | 11.21 | 3.41 | 4.67 | 68.07 | 29.23 | 1.33 |
| DMR WNC NY2143 | 12.07 | 2.33 | 4.61 | 68.41 | 24.00 | 1.20 |

**Table 1.** *Cont.*

| PEDIGREE | Protein (%) | Fat (%) | Sugar (%) | Starch (%) | 100-K wt. (g) | Specific Gravity (g/cm$^3$) |
|---|---|---|---|---|---|---|
| DMR WNC NY2144 | 11.66 | 3.18 | 4.25 | 70.87 | 27.91 | 1.21 |
| DMR WNC NY 2474 | 11.54 | 2.90 | 4.36 | 68.88 | 33.37 | 1.19 |
| DMR WNC NY 2139 | 11.59 | 3.11 | 4.56 | 71.25 | 26.26 | 1.19 |
| DMR WNC NY 2145 | 11.93 | 3.32 | 5.77 | 69.82 | 26.31 | 1.19 |
| DMR WNC NY 2163 | 11.66 | 3.21 | 4.69 | 74.09 | 22.18 | 1.11 |
| DMR WNC NY 2225 | 11.52 | 2.88 | 3.76 | 73.38 | 29.19 | 1.22 |

### 2.2. Preliminary Analysis

The samples were oven dried at 90 °C to reduce the level of moisture of the grains to meet the accuracy of the results. The kernels were ground to powder by course and fine grinding using a pestle and mortar and finally kept in desiccators for analysis of various nutritional quality parameters.

### 2.3. Estimation of Protein, Moisture, Sugar, Starch, 100-Kernel Weight (100-Kernel wt.), Specific Gravity, and Fat Concentration

Protein concentration was determined by the micro-Kjeldahl method of AOAC [22]. In this method, the de-fatted samples were digested until the solution became colorless. Further distillation and titration were conducted by using 8 mL NaOH and 0.02 N HCl. The moisture level of the grains was determined by the oven drying method OAC 934.01 by drying at 135 °C for 2 h [22]. Total sugars were estimated according to the method of Nelson–Somogyi [23]. Starch concentration was determined according to the method of Clegg [24] using Anthrone reagent. After extraction of starch with perchloric acid, it was further hydrolyzed (in an acidic medium) into glucose, which formed a green color compound on reaction with Anthrone reagent. The optical density was recorded against blank at 620 nm. To calculate the 100-kernel weight, one hundred maize kernels were counted manually, and then these were weighed by an electronic weighing balance with 0.01 g accuracy. The specific density of the kernel was determined by the method of Sangamithra [25]. Fat concentration was estimated according to the method of AOAC by using the solvent extractor system [22]. In this method, the fat concentration of the ground powder was extracted at 40–60 °C using non-polar solvent petroleum ether.

### 2.4. Statistical Analysis

#### 2.4.1. Analysis of Variance (ANOVA)

ANOVA was performed to study the significance of the genotypic differences and conducted using Statistical Package for the Social Sciences (SPSS) software. All the biochemical evaluations were conducted in duplicate.

#### 2.4.2. Univariate and Multivariate Statistics

The results were expressed as univariate and multivariate statistics. Scott–Knott correlations at significance levels of 1% and 5% and hierarchical cluster analysis (HCA) based on the squared Euclidean distance using Ward's method were performed using Statistical Analysis Software (SAS 9.2 English). Factor analysis for variability component loading was conducted using Statistical Package for the Social Sciences (SPSS) software. The principal component analysis (PCA) method was used as the extraction. The loading plot was prepared using SPSS. The Pearson correlation coefficient $|r|$ among 63 maize inbred lines was calculated by the formula given below:

$$r = \frac{\sum XY - \frac{(\sum X)(\sum Y)}{n}}{\sqrt{\left(\sum X^2 - \frac{(\sum X)^2}{n}\right)\left(\sum Y^2 - \frac{(\sum Y)^2}{n}\right)}} \tag{1}$$

where $X$ and $Y$ are the variables and $n$ is the total number of samples used in the study.

## 3. Results

### 3.1. Variability Analysis and Factor Analysis

The ANOVA indicated that the inbred lines differed significantly for all the nutritional quality traits of maize kernel (Table 2), and a wide variability among these quality traits was also observed.

**Table 2.** Mean, standard error, and F-ratio of nutritional quality attributes of 63 maize inbred lines for variability analysis.

| Variables | N | Mean | F-Ratio | Minimum | Maximum |
|---|---|---|---|---|---|
| Protein | 63 | 11.34 ± 0.15 * | 13.61 | 8.83 | 15.54 |
| Sugar | 63 | 3.54 ± 0.08 * | 22.07 | 3.01 | 5.37 |
| Starch | 63 | 70.46 ± 0.04 ** | 21.37 | 67.43 | 75.31 |
| Fat | 63 | 2.81 ± 0.25 * | 4.11 | 2.04 | 4.47 |
| Moisture | 63 | 9.11 ± 2.0 * | 2.80 | 9.16 | 10.49 |
| Specific gravity | 63 | 1.25 ± 0.11 | 12.89 | 0.96 | 1.90 |
| 100-K wt. | 63 | 25.08 ± 0.23 | 206.72 | 9.14 | 36.11 |

* Significant at the 0.05 probability level; ** significant at the 0.01 probability level.

The protein concentration in the maize kernels varied from 8.83 to 15.54%. The highest protein concentration was observed in the DMR WNC NY 403 inbred line (Table 2). The sugar concentration ranged from 3.01 to 5.77%, and the highest sugar concentration was observed in the DMR WNC NY 2417 inbred line. The starch concentration of the maize kernels varied from 67.43 to 75.31%. The highest starch concentration was observed in the DMR WNC NY 2408 inbred line. Meanwhile, the fat concentration varied from 2.04 to 4.47%. The highest fat concentration was observed in the DMR WNC NY 400 inbred line. The 100-kernel wt. varied from 9.14 to 36.11 gm, the highest being exhibited by DMR WNC NY 2234, and specific gravity varied from 0.96 to 1.90 g/cm$^3$. Specific gravity data were rightly skewed to their means and contributed towards a high specific gravity. The moisture level of all 63 inbred lines chosen for this study was within acceptable limits.

Factor analysis indicated that the variability among the present inbred lines is mainly contributed by protein and starch concentrations, followed by 100-kernel wt. (Table 3). These three components were extracted based upon principal component analysis, which showed that these alone contribute 68.58% towards kernel variability. The loading plot (Figure 1) revealed that protein, starch concentration, and 100-kernel wt. strongly influence the variability as their loading is close to 1.0. However, the variability does not affect sugar, oil, and specific gravity as their values are below 0.5, i.e., less than 1.0, and as we know, loadings close to −1 or 1 strongly influence the variable, and higher loadings either positively or negatively indicate that the particular variable has a strong effect on the principal component.

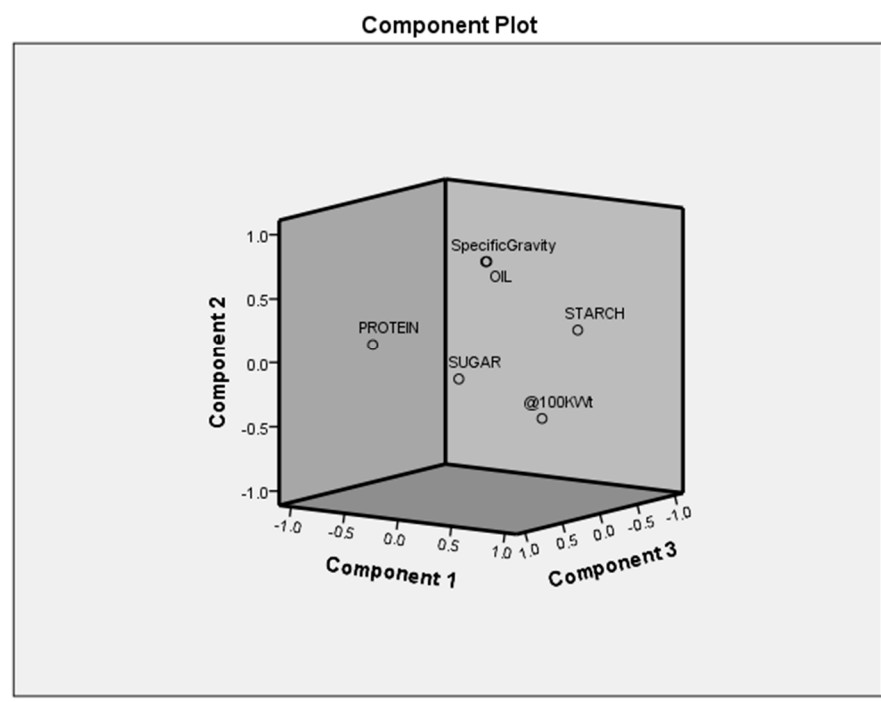

**Figure 1.** Loading plot of nutritional quality attributes of 63 maize inbred lines for variability analysis.

**Table 3.** Component matrix, communities, and total variance of nutritional quality attributes of 63 maize inbred lines for variability analysis. (**a**) Component matrix. (**b**) Communities. (**c**) Total variance explained.

|  | Component | | |
|---|---|---|---|
|  | **1** | **2** | **3** |
| Protein | −0.750 | 0.073 | 0.376 |
| Starch | 0.714 | 0.243 | −0.272 |
| 100-Kernel wt. | 0.688 | −0.391 | 0.168 |
| Fat | 0.118 | 0.770 | 0.107 |
| Specific Gravity | 0.083 | 0.758 | 0.042 |
| Sugar | 0.398 | −0.012 | 0.865 |

|  | **Initial** | **Extraction** |
|---|---|---|
| Protein | 1.000 | 0.709 |
| Oil | 1.000 | 0.618 |
| Sugar | 1.000 | 0.907 |
| Starch | 1.000 | 0.643 |
| 100-Kernel wt. | 1.000 | 0.655 |
| Specific Gravity | 1.000 | 0.583 |

| Component | Initial Eigenvalues | | | Extraction Sums of Squared Loadings | | |
|---|---|---|---|---|---|---|
|  | **Total** | **Variance (%)** | **Cumulative (%)** | **Total** | **Variance (%)** | **Cumulative (%)** |
| 1 | 1.725 | 28.750 | 28.750 | 1.725 | 28.750 | 28.750 |
| 2 | 1.384 | 23.074 | 51.824 | 1.384 | 23.074 | 51.824 |
| 3 | 1.006 | 16.762 | 68.586 | 1.006 | 16.762 | 68.586 |
| 4 | 0.745 | 12.420 | 81.006 | | | |
| 5 | 0.627 | 10.452 | 91.458 | | | |
| 6 | 0.513 | 8.542 | 100.000 | | | |

Extraction method: principal component analysis. Extraction was performed by the principal component analysis method.

### 3.2. Correlation Analysis

Scott–Knott correlation coefficients among various traits such as protein, fat, starch, sugar, 100-kernel wt., and specific gravity of the grains are provided in Table 4. These results reveal that the protein concentration exhibited a significant negative correlation with starch and 100-kernel weight. Contrary to this, the fat concentration showed a significant low positive relationship with specific gravity. Sugar and starch possess a non-significant positive correlation with all the traits under study, except protein.

**Table 4.** Coefficients of nutritional quality attributes of 63 maize inbred lines.

| Parameters | Sugar | Starch | 100-K wt. | Specific Gravity | |
|---|---|---|---|---|---|
| *Protein* | −0.049<br>P = 0.702 | −0.048<br>P = 0.711 | −0.392 **<br>P = 0.001 | −0.351 **<br>P = 0.005 | 0.031<br>P = 0.811 |
| Fat | | 0.076<br>P = 0.553 | 0.129<br>P = 0.315 | −0.140<br>P = 0.272 | 0.283 *<br>P = 0.025 |
| Sugar | | | 0.101<br>P = 0.429 | 0.247<br>P = 0.051 | 0.016<br>P = 0.900 |
| Starch | | | | 0.227<br>P = 0.074 | 0.149<br>P = 0.245 |
| 100-K wt. | | | | | −0.091<br>P = 0.479 |

** Correlation is significant at the 0.01 level; * correlation is significant at the 0.05 level.

### 3.3. Genetic Distance Measurement and Hierarchical Cluster Analysis

The genetic distance relationship of 63 maize inbred lines, depicted by the squared Euclidean distance based on Ward's method of hierarchical clustering, was obtained based on nutrient composition (Figure 2) data. Cluster analysis was used to reveal the association between the inbred lines used in the present study. Cluster analysis provides various clustering algorithms such as "sequential hierarchical and neighbor clustering". Due to the great heterogeneity within groups, hierarchical cluster analysis can be used to cluster maize inbred lines according to their differences and similarities to further investigate chemical compositional relationships between them. A total of two major clusters are formed at a distance of 25, having 26 and 37 inbred lines in cluster I and cluster II, respectively. On further reducing the distance to 9, three clusters were formed in which cluster I remained unaffected, while cluster II was split into two sub-clusters, viz., C II-SC 1 and C II-SC 2, having 7 and 30 maize inbred lines, respectively. On further reducing the distance to 6, cluster I was split into two sub-clusters, viz., C I-SC 1 and C I-SC 2, having 6 and 20 members, respectively. DMR WNC NY 397 and DMR WNC NY 404 are the farthest apart, having major differences in their protein, fat, starch, specific gravity, 100-kernel wt., and sugar concentration, followed by DMR WNC NY 2436 and DMR WNC NY 2394, DMR WNC NY 2212 and DMR WNC NY 2430, DMR WNC NY 396 and DMR WNC NY 2415, DMR WNC NY 404 and DMR WNC NY 2144, and DMR WNC NY403 and DMR WNC NY 2115. This variability can be exploited in crop improvement programs, particularly for grain quality traits.

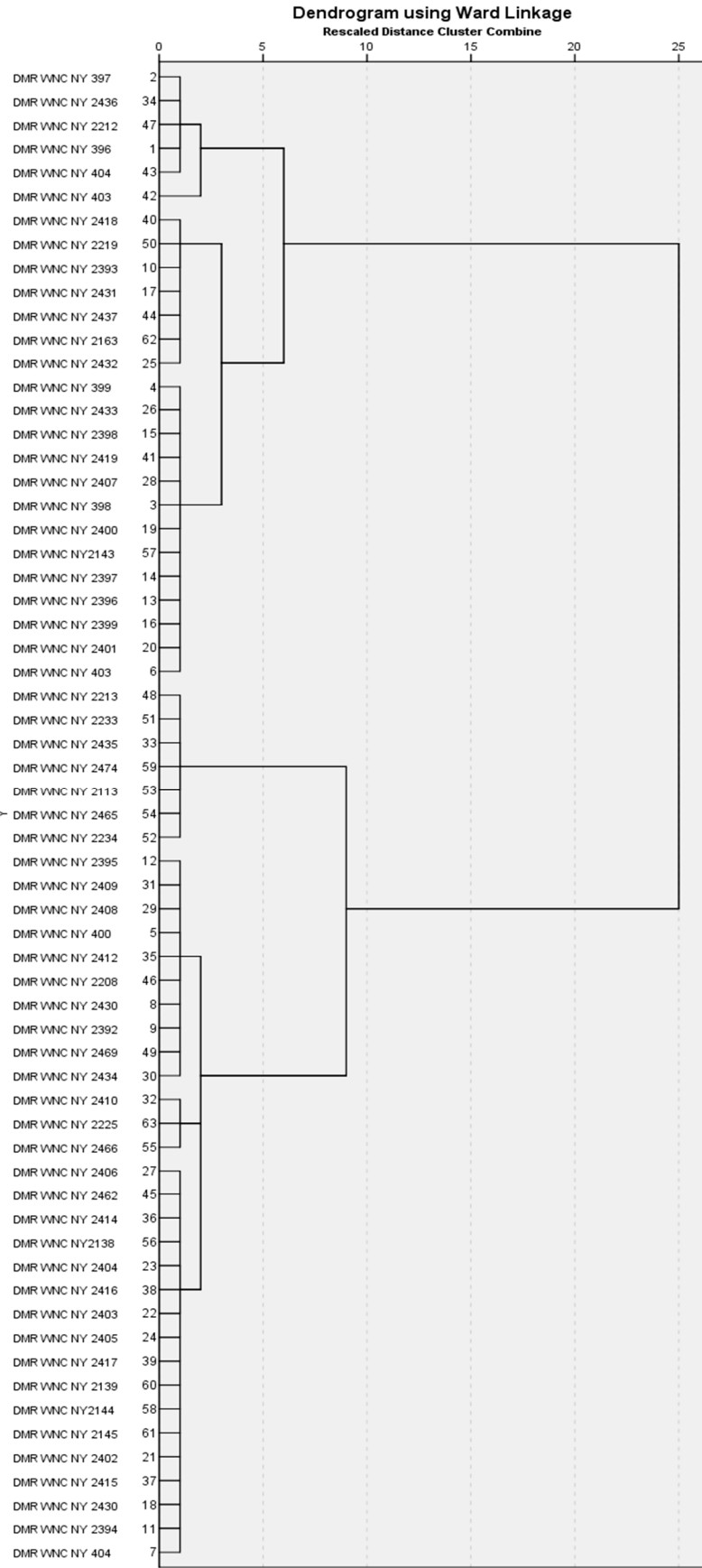

**Figure 2.** Depicting hierarchical clustering of 63 maize inbred lines by squared Euclidean distance based on Ward's method of hierarchical clustering based on their nutritional composition differences.

## 4. Discussion

Maize kernels contain essential components for plant growth and reproduction [23]. However, the nutritional quality of maize is determined by its genetic background, and hence the inbred varieties with a high concentration of proteins, sugars, and other nutritional attributes are desired to combat malnutrition, particularly in sub-Saharan Africa, Southeast Asia, and Latin America [3].

A significant negative correlation between the protein concentration and starch and 100-kernel wt. indicated that an increase in the protein concentration will down-regulate starch and 100-kernel wt. This might be because protein and starch are both mostly present in the endosperm part of the kernel [26]. Although the endosperm contains lower amounts (8%) of protein than the germ (18.4%), it provides the highest quantity, i.e., 78% as compared to the 18% provided by the kernel germ [27]. Contrary to this, the fat concentration showed a significant low positive relationship with specific gravity, indicating that the specific gravity of maize may be increased with an increase in the fat concentration. Therefore, selecting for high fat and specific gravity altogether will not cause any difficulty during crop improvement programs. Maize starch accounts for about 72% of the dry weight of kernels, and it is located in the endosperm, while sugars such as sucrose, maltose, glucose, fructose, and raffinose constitute about 1–3% of the total dry weight of kernels [28]. The major (75%) portion of these free sugars is located in the germ, with the endosperm constituting only 25% of the total sugars [28]. A negative low correlation between protein and fat revealed that breeding efforts cannot be applied for increasing both energy-rich sources simultaneously, and genetic manipulations to increase one may affect the other. Another result of this study is a non-significant positive correlation of sugar and starch with all the traits under study, except protein, as starch contributes mainly to the grain filling characteristics of maize, hence increasing the 100-kernel wt. and, simultaneously, the density of the kernel. Specific gravity was reported to enhance the viability of the grains through an increase in their quality in soybean [29].

Muhammad Saleem et al. [30] and Chaudhary et al. [21] found a positive correlation between protein and sugars. The protein in maize kernels has been studied extensively because it forms the core of the quality of maize. This might be the reason for the above results because sugars are mainly confined to the germ as discussed earlier as well. Our results agree with some other previous findings of Willmot et al. [31]; Clark et al. [32]; Dudley et al. [33]; Liu et al. [34]; Zhang et al. [35]; and Chaudhary et al. [21]. A negative correlation between protein and moisture concentrations indicates an increased moisture level of the grains would decrease the quality of the grain. Therefore, a balanced moisture level is required for improving the quality as well as the quantity of grains. Chaudhary et al. [21] postulated that in maize, the germ accounts for 8–10% of the total grain weight and may contribute 15–20% of total protein, whereas the endosperm accounts for 80–85% and contributes approximately 80% of protein. Panthee et al. [36] found an inverse relationship between protein and fat concentrations and hence postulated that it will be difficult to improve both attributes simultaneously. Generally, it was observed by many researchers that the protein concentration is negatively and positively correlated with starch and fat concentrations [31,32,34,35]. Similarly, an inverse relation was found between starch and fat [37]. An increase in fat, which might have originated from a bigger germ size, could adversely affect the endosperm volume, thus indirectly contributing towards a lower starch concentration. High-fat maize is usually maize with a higher germ size and a lower endosperm if compared to normal maize genotypes [21]. Few researchers reported that there would be no serious barrier in selecting for both high-protein and high-fat in maize [38,39]. Okporie and Obi [39] and Okporie Oselebe [40] found that sugar also has a very low non-significant positive correlation with starch, which may be attributed to the source and sink relationship. Generally, a high sugar concentration has comparatively low starch synthase activity that may be due to the high Km value of enzymes. Here, both might be there in the equilibrium phase.

Genetic distance analysis presented two main clusters having 26 and 27 clusters, providing the base for selecting parents at a farthest distance in order to attain nutritional variability.

Various strategies for attaining nutritional security can be adopted which are feasible and sustainable as well. Bio-fortification is one of the most important strategies because it is rural-based, where 70% of the resource-poor live, and is cost-effective and sustainable too. Maize provides about 15% of the world's protein (essential amino acids) and 20% of the world's calories [5]. A maize inbred line rich in protein and sugar concentrations has a positive correlation between these nutrients and is highly important for proper human nutrition, growth, health, and immunity and for combating malnutrition.

The outcomes of the present study will help in designing breeding strategies for developing nutritionally improved maize hybrids for the nutritional security of India and the world, and this can answer many of the nutritional quality issues. The authors strongly believe that along with these strategies, bioavailability studies should be carried out to confirm that the targeted nutrient is being readily absorbed in the body and can fulfill the recommended dietary allowance (RDA) because the nutritional quality of the "resource-poor" is much more important than that of the quantity. This approach may help to take a step forward to improve the livelihood security of a nation.

## 5. Conclusions

The results of the present study reveal a wide variability among protein, starch concentration, and 100-kernel weight in maize inbred lines, and these three contribute to 68.58% of the kernel variability. The inbred lines were proposed as donors for the development of high cultivars for their respective traits, viz., high protein (DMR WNC NY 403 and DMR WNC NY 404), starch concentration (DMR WNC NY 2163, DMR WNC NY 2219, DMR WNC NY 2234, DMR WNC NY 2408, DMR WNC NY 2437, and DMR WNC NY 2466), 100-kernel wt. (DMR WNC NY 2113, DMR WNC NY 2213, DMR WNC NY 2233, DMR WNC NY 2234, DMR WNC NY 2414, DMR WNC NY 2435, DMR WNC NY 2465, and DMR WNC NY 2474), sugar (DMR WNC NY 2417), and specific gravity (DMR WNC NY 2418). Nutritional components in maize inbred lines are also highly correlated, and an alteration in one may positively or negatively affect the other. Although all three were the principal components of variability, an increase in protein will lower the starch concentration. Further, an increase in the weight of the kernels might increase the starch concentration rather than lower the protein concentration of the grain. The development of high-protein maize will affect the grain yield to some extent. However, selecting for high fat would increase the specific gravity which in turn enhances the viability of the grain, providing a wide base for maize hybridization programs.

DMR WNC NY 403 and DMR WNC NY 404 are proposed as high-protein, low-sugar, and low-starch materials, DMR WNC NY 2163, DMR WNC NY 2219, DMR WNC NY 2234, DMR WNC NY 2408, DMR WNC NY 2437, and DMR WNC NY 2466 as high-starch and low-protein materials, and DMR WNC NY 2418 for its high specific gravity and fat level. DMR WNC NY 2234 is proposed as a promising material for its starch concentration and 100-K wt., and DMR WNC NY 2163 and DMR WNC NY 2219 are proposed for their protein, sugar, starch, and specific gravity, only needing to improve their 100-K wt. and fat concentration. Further, DMR WNC NY 2113, DMR WNC NY 2408, DMR WNC NY 2417, and DMR WNC NY 2437 are proposed as excellent materials for their protein, sugar, starch, 100-K wt., and specific gravity, only needing to improve their fat content. Next, DMR WNC NY 2213, DMR WNC NY 2465, and DMR WNC NY 2466 are found as promising for all other traits than protein and fat concentration, whereas DMR WNC NY 2414 and DMR WNC NY 2435 need improvement in their fat and starch concentrations only.

Among the various inbred lines, DMR WNC NY 397 and DMR WNC NY 404, followed by DMR WNC NY 2436 and DMR WNC NY 2394, DMR WNC NY 2212 and DMR WNC NY 2430, DMR WNC NY 396 and DMR WNC NY 2415, DMR WNC NY 404 and DMR WNC NY 2144, and DMR WNC NY403 and DMR WNC NY 2115, were genetically the farthest

apart inbred lines, having major differences in their protein, fat, starch, 100-kernel weight, specific gravity, and sugar concentration, and arising from two different clusters as well. In brief, there is a high variability for three major nutritional traits as identified by factor analysis; therefore, these inbred lines can be used as potential donors of the respective traits and would be beneficial to be proposed in breeding programs, as combinations of these lines and their crosses would result in hybrids or genetic variability with high values in one or many traits.

The adoption of maize inbred lines possessing higher proteins and amino acids would result in a significant decrease in malnutrition. Development and consumption of nutrient-rich maize varieties would help in preventing malnutrition and in achieving nutritional security more holistically.

**Author Contributions:** Conceptualization, S.L.; investigation and writing of the original draft of the manuscript, S.L.; methodology, S.L., D.P.C., J.C.S., and Z.A.D.; reviewing and editing of the manuscript, S.R.; editing and formal analysis, S.H., H.E.E., and R.Z.S.; funding acquisition, H.E.E. All authors have read and agreed to the published version of the manuscript.

**Funding:** This research was funded by Allcosmos Industries Sdn. Bhd. Arif Efektif Sdn. Bhd., Malaysia with grant Ns. RJ130000.7609.4C187 and RJ130000.7344.4B200.

**Institutional Review Board Statement:** Not applicable.

**Informed Consent Statement:** Not applicable.

**Data Availability Statement:** All the data is available in the manuscript.

**Conflicts of Interest:** The authors declare no conflict of interest.

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
