# Peer review of "Analysis of Nutritional Quality Attributes and Their Inter-Relationship in Maize Inbred Lines for Sustainable Livelihood"

_sustainability, doi:10.3390/su13116137_

Round 1

Reviewer 1 Report

  • Quite interesting research since it gave results about the variability of maize about important nutrient components.

Abstract.

  • LINES 25-28. Authors wrote “Factor analysis revealed that Protein content, starch, and 100 kernel weight, the three major components alone comprises 68.58% of the kernel composition”. I think that the last part in wrong e.g. “Factor analysis revealed that Protein content, starch, and 100 kernel weight, the three major components alone comprises 68.58% of the kernel composition”. Maybe authors mean “58% of the kernel variability?”
  • I propose to authors to enrich the abstract with results/ conclusions e.g. at the end of the abstract to propose which lines could contribute in genetic variability with high/low protein, starch, fat etc in a breeding program. Moreover, to propose possible combination of lines that their crosses (in a breeding program) would result in genetic variability with simultaneous high values for important characteristics e.g. protein, starch, or fat etc, or even for characteristic that correlated negatively.

Introduction

  • The introduction is quiet informative but authors should reduce the general part e.g. lines 37-86 and increase the importance of the maize in nutrition e.g. lines 87-102.

  • Please make more precise the aim of the study e.g. lines 103-111.

Please consider my proposal for the aim of the study.

The present research work was undertaken to study the variability, the correlation and the in­terrelationship between nutrient components of 63 different varie­ties of maize inbreds. Also, the genetic distance between 63 different varie­ties of maize inbreds was studied in order to be selected those that having the potential of using as donor parents for their respective traits for the development of maize cultivars with enhanced protein, sugar, and minerals, with a possible contribution in decreasing in malnutrition.”

Material and method

2.1 Plant Materials, line 113-118. Authors should present information about the experimental design (date of establishment in the field, experimental design, size of plot, fertilization etc) and not results of the experiments such as they did –lines 115-117 “The inbred plants' height ranged from 0.9-1.3 m, with low to medium cob placement. The anthesis occurred in 70­75 days and the final maturity was attained in 125-130 days.”

  • İn table 1. Lines119. Authors should present statistical diferrences of the maize lines for protein, fat etc.
  • In 2.4-lines (143-153) Authors should present separately the ANOVA and the mutlivariate analysis.
  • The results are written very simplified.
  • Regarding correlation 3.2 lines 191-204. The correlation coefficient takes absolute values 0 to 1. So if r=0.283, is considered a low positive correlation, and even if it is significant is not so important. So please change lines 191-204 and the part of the discussion that referred to correlation according to this comment.

Discussion

-Author could reduce the general part of the discussion (lines 227-251)

-Author should rewrite the part of the correlation (lines 274-303) according to the above comments. “Regarding correlation”.

- Authors should propose maize lines that could be used as donor parents for their respective traits for the development of: 1. maize cultivars with enhanced protein, sugar, and minerals, or 2. maize cultivars that combine high values in many traits (protein, fat, sugar, starch, 100k wt, specific gravity). Also, it would be beneficial to be proposed combination of lines that their crosses would result in hybrids or genetic variability with high values in one or many traits (protein, fat, sugar, starch, 100k wt, specific gravity).

-Author should refer a part of the discussion for the Dendrogram or the groups of the maize lines.

Conclusion

-Authors should rewrite the part of the correlation.

-Should focus on the important findings of the variance of many traits (protein, fat, sugar, starch, 100k wt, specific gravity)

- Authors should propose which lines could contribute in genetic variability with high/low protein, starch, fat etc in a breeding program. Moreover, to propose possible combination of lines that their crosses (in a breeding program) would result in genetic variability with simultaneous high values for important characteristics e.g. protein, starch, or fat etc, or even for characteristic that correlated negatively

Minor changed

  • The word “major” appears many time in the manuscript. Please replace wherever you can with synonyms like: important, significantly etc.
  • English language and grammar control it is necessary to be improved. I propose many changes in the pdf.

Reviewer 1 Report

Comments and Suggestions for Authors

Quite interesting research since it gave results about the variability of maize about important nutrient components.

Abstract.

  • LINES 25-28. Authors wrote, “Factor analysis revealed that Protein content, starch, and 100 kernel weight, the three major components alone comprises 68.58% of the kernel composition”. I think that the last part is wrong, e.g., “Factor analysis revealed that Protein content, starch, and 100 kernel weight, the three major components alone comprises 68.58% of the kernel composition”. Maybe authors mean “68% of the kernel variability?”

Authors’ response: Yes, very right. We have changed the same in the text as well. Thank you for your valuable suggestion.

  • I propose to authors to enrich the abstract with results/ conclusions, e.g., at the end of the abstract to propose which lines could contribute to genetic variability with high/low protein, starch, fat, etc., in a breeding program. Moreover, propose a possible combination of lines that their crosses (in a breeding program) would result in genetic variability with simultaneous high values for important characteristics, e.g., protein, starch, or fat, etc., even for characteristics that correlated negatively.

Authors’ response: All the suggestions are incorporated in the text. The abstract is enriched with the potential donors of the traits and parents used in a breeding program. 

Introduction

  • The introduction is quite informative, but authors should reduce the general part, e.g., lines 37-86, and increase the importance of the maize in nutrition, e.g., lines 87-102.

Authors’ response: The same has been done as per suggestion. Kindly refer to the text in the manuscript.  

  • Please make more precise the aim of the study, e.g., lines 103-111.

Please consider my proposal for the aim of the study.

“The present research work was undertaken to study the variability, the correlation, and the in­terrelationship between nutrient components of 63 different varie­ties of maize inbreds. Also, the genetic distance between 63 different varie­ties of maize inbreds was studied to be selected those that having the potential of using as donor parents for their respective traits for the development of maize cultivars with enhanced protein, sugar, and minerals, with a possible contribution in decreasing in malnutrition.”

Authors’ response: The text is replaced by the suggested one. Kindly refer to the text in the manuscript. Your suggestions are really valuable.

Material and method

2.1 Plant Materials, line 113-118. Authors should present information about the experimental design (date of establishment in the field, experimental design, size of plot, fertilization, etc.) and not results of the experiments such as they did –lines 115-117 “The inbred plants' height ranged from 0.9-1.3 m, with low to medium cob placement. The anthesis occurred in 70­75 days, and the final maturity was attained in 125-130 days.”

Authors’ response: The portion is updated as per suggestion as “The inbreds were grown in an augmented block design (ABD) at Winter Nursery Centre, Hyderabad, India having a temperature range of 7 oC to 12 oC, sandy soil, 6.5 pH, low in organic matter, during rabi season. The plants were selfed, and the selfed seeds were used for biochemical evaluation. Details of the pedigree of the inbred line are given in Table 1”.

  • İn table 1. Lines119. Authors should present statistical differences of the maize lines for protein, fat, etc.

Authors’ response: thank you for the suggestion. As we have a separate table for Mean, Standard Error, and F-ratio (Table 2), we did not add to avoid duplicity.

  • In 2.4-lines (143-153), the Authors should present the ANOVA and the multivariate analysis separately.

Authors’ response: The same is divided into two sections as per suggestion. Kindly refer to the text in the manuscript.

  • The results are written very simplified.

Authors’ response: Results are revised now. Hope it meets expectation now!

  • Regarding correlation 3.2 lines 191-204. The correlation coefficient takes absolute values 0 to 1. So if r=0.283, it is considered a low positive correlation, and even if it is significant is not so important. So please change lines 191-204, and the part of the discussion referred to correlation according to this comment.

Authors’ response: The correlation portion is revised now. Kindly refer to the text.

Discussion

-Author could reduce the general part of the discussion (lines 227-251)

Authors’ response: Yes, done as per suggestion. Kindly refer to the text in the manuscript.

According to the above comments, the author should rewrite the part of the correlation (lines 274-303). “Regarding correlation.”

Authors’ response: Yes, it is rewritten now. Please refer to the respective portion in the text.

- Authors should propose maize lines that could be used as donor parents for their respective traits for the development of 1. maize cultivars with enhanced protein, sugar, and minerals, or 2. maize cultivars that combine high values in many traits (protein, fat, sugar, starch, 100k wt, specific gravity)Also, it would be beneficial to be proposed a combination of lines that their crosses would result in hybrids or genetic variability with high values in one or many traits (protein, fat, sugar, starch, 100k wt, specific gravity).

Authors’ response: All the proposed suggestions are incorporated now.

-Author should refer to a part of the discussion for the Dendrogram or the groups of the maize lines.

Authors’ response: Yes, done as per suggestion. Added 2-3 lines for the same.

Conclusion

-Authors should rewrite the part of the correlation.

Authors’ response: The same has been done. Thank you!

-Should focus on the important findings of the variance of many traits (protein, fat, sugar, starch, 100k wt, specific gravity)

Authors’ response: The same has been done. Kindly refer to the section.

- Authors should propose which lines could contribute to genetic variability with high/low protein, starch, fat, etc., in a breeding program. Moreover, to propose a possible combination of lines that their crosses (in a breeding program) would result in genetic variability with simultaneous high values for important characteristics, e.g., protein, starch, or fat, etc., or even for the characteristic that correlated negatively

Authors’ response: The same has been included in the conclusion part to the best.

Minor changed

  • The word “major” appears many times in the manuscript. Please replace wherever you can with synonyms like important, significant, etc.

Authors’ response: The same is replaced by important and other similar words.

  • English language and grammar control are necessary to be improved. I propose many changes in the pdf.

Authors’ response: All the proposed changes are incorporated along with some other grammar control. We are really thankful for your valuable suggestions, as these have improved the manuscript.

Reviewer 2 Report

Highly relevant work on maize to determine which genotypes have differing characteristics, on which to focus future hybrid crosses for increased nutritional content of the grain.

A few minor comments and suggestions:

Line 53: change ‘till’ to ‘until’

Line 114 indicates that the plants were grown in blocks, but not if they are replicated blocks, and how many replications. Did you only have one plant per variety? Maize plants can vary in growth (kernels, too), from plant to plant, depending on roots randomly growing differently and getting different nutrition from the soil.

What were the soil properties? What type of soil? pH, organic matter? I assume that these plants were grown without any fertilizer? Corn growth and grain properties are highly dependent upon soil/ fertilizer conditions. I understand that these are inbreds, but I assume that these conditions would be representative of a typical farming practice in India?

Table 1: Why are some 100K weights rounded, and some not? Please be consistent. Also, are these values displayed on a 0% moisture basis?

Line 136-8: If the authors manually counted out 100 kernels and then weighed them, why were the mass data extrapolated to 100 kernels?

Equation 1: please include a sentence defining X, Y and n.

Line 162+: All of your words of content should be changed to “concentration” or “level”. Content would be total amount per grain or per plant (or even per area); however, % weight is concentration.

Line 169: DMR WNC NY 2234 seems to be a larger font.

Line 187: I do not understand your phrase “… values are below 0.5 i.e. less than 1.0”. Please add a few more words to this sentence to help your readers.

Figure 1: Could these data values be changed to bar-graph type? It’s very hard to determine a three-dimensional point on a two-dimensional page.

Table 4: Why are some Parameters in CAPS, others not, PROTEIN in bold, but others not?

Line 267-9: I do not agree with the hypothesis of ‘Grains with more proteins and sugars will have more metabolic activities and the increased metabolic activities will improve the grain size and hence grain yield…” Maize grain with high protein concentration is typically compact and small. Increasing the protein concentration will not increase yield. As the authors note, protein and starch are inverse, and starch contributes mainly to the grain filling and 100-k weight. The authors also note in their conclusions (L. 333) that ‘…high-protein maize will adversely affect the grain yield…’

Reviewer 2 Report

Comments and Suggestions for Authors

Highly relevant work on maize to determine which genotypes have differing characteristics, on which to focus future hybrid crosses for the increased nutritional content of the grain.

A few minor comments and suggestions:

  • Line 53: change ‘till’ to ‘until’

Authors’ response: Line 53, till is replaced by until

  • Line 114 indicates that the plants were grown in blocks, but not if they are replicated blocks, and how many replications. Did you only have one plant per variety? Maize plants can vary in growth (kernels, too), from plant to plant, depending on roots randomly growing differently and getting different nutrition from the soil.

Authors’ response: Plants were grown in an augmented block design (ABD). It is a fact that maize plants, specially inbred lines, vary greatly in their biological information. Therefore, we took two biological replicates per sample. The selfed seeds were of high purity and, therefore, used for the study. The text is changed as per the suggestion in the manuscript.

  • What were the soil properties? What type of soil? pH, organic matter? I assume that these plants were grown without any fertilizer? Corn growth and grain properties are highly dependent upon soil/ fertilizer conditions. I understand that these are inbreds, but I assume that these conditions would represent typical farming practice in India?

Authors’ response: Yes, it is. The inbreds were grown in augmented blocks design (ABD) at Winter Nursery Centre, Hyderabad, India having a temperature range of 7 oC to 12 oC oC, sandy soil, 6.5 pH, low in organic matter, during rabi season. The plants were selfed, and the selfed seeds were used for biochemical evaluation.

  • Table 1: Why are some 100K weights rounded and some not? Please be consistent. Also, are these values displayed on a 0% moisture basis?

Authors’ response: Table 1 decimal factor is kept consistent now as per suggestion. All the studies were done having a moisture level of 9.16-10.49, as displayed in table 2.

  • Line 136-8: If the authors manually counted out 100 kernels and then weighed them, why were the mass data extrapolated to 100 kernels?

Authors’ response: The line…..and finally extrapolating this mass to 100 kernels….has been deleted from the text as the weight is represented on an actual basis.

  • Equation 1: please include a sentence defining X, Y, and n.

Authors’ response: added a line……” Where X and Y are the variables, and N represents the total no. of the samples used in the study”.

  • Line 162+: All of your words of content should be changed to “concentration” or “level.” Content would be the total amount per grain or per plant (or even per area); however, % weight is concentration.

Authors’ response: The same is updated in the text. All the “content” has been replaced by concentration or level.

  • Line 169: DMR WNC NY 2234 seems to be a larger font.

Authors’ response: Formatting done as per the rest of the document.

  • Line 187: I do not understand your phrase “… values are below 0.5, i.e., less than 1.0”. Please add a few more words to this sentence to help your readers.

Authors’ response: Added more information in the text “As we know that loadings close to -1 or 1 strongly influence the variable and higher loadings either positive or negative indicates that the particular variable has a strong effect on the principal component”.

  • Figure 1: Could these data values be changed to bar-graph type? It’s very hard to determine a three-dimensional point on a two-dimensional page.

Authors’ response: we understand the concern, but the loading plot cannot be changed to a bar graph. The same can be interpreted from PCA (factor analysis) itself for more explanation.

  • Table 4: Why are some Parameters in CAPS, others not, PROTEIN in bold, but others not?

Authors’ response: All the parameters are made consistent as per the reviewer’s comment.

  • Line 267-9: I do not agree with the hypothesis of ‘Grains with more proteins and sugars will have more metabolic activities and the increased metabolic activities will improve the grain size and hence grain yield…” Maize grain with high protein concentration is typically compact and small. Increasing the protein concentration will not increase yield. As the authors note, protein and starch are inverses, and starch contributes mainly to the grain filling and 100-k weight. In their conclusions (L. 333), the authors also note that ‘…high-protein maize will adversely affect the grain yield….’

Authors’ response: During revision, this portion is deleted, and we have rewritten the correlation part and updated in abstract, results, discussion and conclusion. Hope it meets the expectation now. We are really thankful for your valuable suggestions, as these have improved the manuscript.

Reviewer 3 Report

Lots of formatting issues

Introduction is unnecessary lengthy and many repetition and unrelated topics.

The research results were presented poorly

Many repetitions of the same findings throughout the results/discussion/conclusion

Line 25: 100 kernel weight (9.14-36.11g/cm3). Why is the unit g/cm3??

Lines 25-27: “Factor analysis revealed that 25 Protein content, starch, and 100 kernel weight, the three major components alone comprises 68.58% 26 of the kernel composition.” Is 100 kernel weight a “kernel composition”?

Table 1: be consistent with the use of decimal points. 100 kernel weight unit (Gm)??

Line 145: two replication is low

Table 2: format of data is not consistent. Why is f-ratio of 100 kW so high??

Where has Figure 1 been referred in the text?

Lines 198-202: If these correlations are not significant, is it worth to mention? One can clearly see what has not been significant from Table 4. Even the significant correlations were not strong, a coefficient of 0.283 is very low.

Table 4: what is the @ sing before the 100KWt for?

Reviewer 3 Report

Comments and Suggestions for Authors

  • Lots of formatting issues

Authors’ response: proper formatting is done to remove the same. It arose due to windows incompatibility.

  • The introduction is unnecessarily lengthy and many repetitions and unrelated topics.

Authors’ response: The introduction is made precise as per suggestion. Many of the unnecessary portions are deleted. Thank you for your valuable comment and for improving the paper.

  • The research results were presented poorly.

Authors’ response: The results are revised as per suggestion, and I hope it works now.

  • Many repetitions of the same findings throughout the results/discussion/conclusion

Authors’ response: All the repetitions are removed now. Kindly refer to the text.

  • Line 25: 100 kernel weight (9.14-36.11g/cm3). Why is the unit g/cm3??

Authors’ response: It was a typographical mistake, and G/cm3 is converted to gm, a unit of 100 kernel wt.

  • Lines 25-27: “Factor analysis revealed that 25 Protein content, starch, and 100 kernel weight, the three major components alone comprises 68.58% 26 of the kernel composition.” Is 100 kernel weight a “kernel composition”?

Authors’ response: The “kernel composition” for the respective text has been changed to “Kernel variability.” Thank you for the suggestion.

  • Table 1: be consistent with the use of decimal points. 100 kernel weight unit (Gm)??

Authors’ response: In Table 1, two decimal points are kept consistent now. Gm is the gram, the unit of 100 kernel wt.

Line 145: two replication is low

Authors’ response: The seeds undertaken for the study were selfed and of high purity. Therefore, we took two replications.

  • Table 2: format of data is not consistent. Why is the f-ratio of 100 kW so high??

Authors’ response: Formatting of data is done again. This is arising due to windows incompatibility. F-ratio is corrected as 206.72 in place of 2067.20.

  • Where has Figure 1 been referred to in the text?

Authors’ response: Figure 1 is referred to in the text at line no. 185.

  • Lines 198-202: If these correlations are not significant, is it worth mentioning? One can clearly see what has not been significant from Table 4. Even the significant correlations were not strong; a coefficient of 0.283 is very low.

Authors’ response: The correlation portion is revised again. Kindly refer to the text. Thank you for the suggestion.

  • Table 4: what is the @ sing before the 100KWt for?

Authors’ response: @ sign is removed from table no. 4. We are really thankful for your valuable suggestions as these have improved the manuscript a lot.

Round 2

Reviewer 1 Report

Ι am glad to read that the majority of proposed changes had been incorporated into the manuscript.

- Lines 131-137. Authors should write information about the experimental plot e.g. size/dimension

-Lines 235-257. I would like authors to study if the different groups of inbred lines characterized from different characteristics e.g. high protein and low starch etc.

- Lines 365-374. I would like author to study if there are a few inbred lines that combined high values to many (of the 6 characteristics) of table 2, in order to propose as promising/excellent material that need to improved only to 1-2 characteristics. If authors find such inbred lines could refer also in the conclusions.

- Authors should read carefully the manuscript in order to improve its cohesion.

Author Response

Reviewer 1 Round 2 Report

Comments and Suggestions for Authors

Ι am glad to read that the majority of proposed changes had been incorporated into the manuscript.

Authors response: The reviewer’s comments and suggestions were highly useful and helped in the improvement of the paper

  • Lines 131-137. Authors should write information about the experimental plot, e.g., size/dimension

Authors response: The following text has been added at Lines 131-137

 4 rows per inbred at 60 cm spacing at a length of 3 m

  • Lines 235-257. I would like authors to study if the different groups of inbred lines are characterized from different characteristics, e.g., high protein and low starch, etc.

Authors response: Agreed. The suggestion has been incorporated. Kindly refer to the discussion part.

  • Lines 365-374. I would like the author to study if there are a few inbred lines that combined high values to many (of the 6 characteristics) of table 2, in order to propose as promising/excellent material that needs to improve only to 1-2 characteristics. If authors find such inbred lines could also refer to the conclusions.

Authors response: The authors thank the reviewer for such a valuable suggestion. The suggestion has been incorporated. Kindly refer to the conclusion part of the manuscript.

  • Authors should read the manuscript carefully in order to improve its cohesion.

Authors response: We have improved the manuscript to the best of our knowledge and thankful to the reviewer for valuable inputs.
